

# Estimating summer sea ice extent in the Weddell Sea during the early nineteenth century

Eleanor Love[1], Grant R. Bigg[1]

[1]Department of Geography University of Sheffield, Winter Street, Sheffield, S10 2TN, U.K.

*Correspondence to*: Grant R. Bigg (grant.bigg@sheffield.ac.uk)

**Abstract.** Over the past three decades, discordant trends in sea ice extent have been observed between the Arctic and Antarctic regions. Arctic sea ice extent has been characterised by a rapid decline, whereas Antarctic sea ice extent, while highly variable inter-annually, has tended to increase. Climate models have so far failed to capture these trends. Coupled with the limited pre-1970 sea ice dataset, this poses a significant challenge to quantifying the mechanisms

responsible for driving such trends. However, historical records from early Antarctic expeditions contain a wealth of information regarding the nature and concentration of sea ice. Such records have been under-utilised, and their analysis may enhance our understanding of recent Antarctic sea ice variability. For the purpose of this study, 9 records from 8 Antarctic expeditions have been examined. Summer sea ice positions recorded during 1820-1843 have been compared to satellite observations from 1987-2017, as well as historical data for the period 1897-1917. Through analysis of these

three time series, estimations for summer sea ice extent in the Weddell Sea, during the early nineteenth century have been produced. The key findings of this study indicate a nineteenth century average core summer northernmost sea ice latitude in much of the Weddell Sea that was further north than during the modern era, with nineteenth century February having significantly more sea ice by all measures. However, late summer sea ice was most extensive in the early years of the twentieth century.



## 1 Introduction

During the last three decades, unique trends in sea ice extent (SIE) have been observed around the Antarctic continent. Specifically, pan-Antarctic SIE has been characterised by a weak and insignificant positive trend from 1978-2022, with record winter maxima observed in 2012, 2013 and 2014 (Turner and Comiso, 2017; Landrum et al., 2017) but record spring-summer minima observed in 2016 and 2017 (Parkinson and DiGirolamo, 2021). Particularly large increases have

been observed in the Ross Sea in the winter and the Weddell Sea in the summer, but significant declines in extent in the Bellingshausen Sea in summer and the Weddell Sea in winter. This tendency for regional change is reflected in the significant interannual variability seen in the otherwise weak hemispheric trend (Fig. 1).

These trends have persisted in contrast to expectations of sea ice decline in a warming world (Polvani and Smith, 2013; Parkinson and DiGirolamo, 2016; Riihelä et al., 2021), and in stark contrast to those observed in the Northern

Hemisphere (Yu et al., 2017; Parkinson and DiGirolamo, 2021). Arctic SIE has exhibited a rapid decline since the late 1970s, with a net loss of approximately 50% of summer sea ice throughout the region (Turner and Comiso, 2017). This decline can be largely attributed to global warming trends and the loss of significant amounts of multi-year sea ice (Wadhams, 2012; Herzfeld et al., 2015). In addition, complex positive feedback mechanisms have amplified surface warming in the Arctic region, giving rise to the term 'Arctic Amplification' (Chen et al., 2019). Fig. 1 presents SIE

trends for the Arctic and Antarctic regions, and demonstrates the discordance between the two hemispheres.

The Polar Regions' response to climate change is critical in defining the future of the global climate. Sea ice may play a vital role in modulating climate on a global scale by controlling climatically influential mechanisms (Morioka et al., 2017; Shao and Ke, 2015; Yu et al., 2017). Specifically, sea ice formation controls ocean-atmosphere exchanges and influences deep-water formation and surface albedo (Yuan et al., 2017; Parmentier et al., 2017). Quantifying the

response of sea ice to climate change is very important; thus, the unique trends in Antarctic SIE have sparked significant interest within the scientific community. The fundamental question remains, are these trends simply the result of natural variability, or of external driving mechanisms, such as anthropogenic forcing? In particular, explaining the greater variability in Antarctic SIE trends relative to those observed in the Arctic, quantifying the role of anthropogenic forcing in driving such trends, and establishing whether the recent decline in Antarctic SIE marks a long-

term shift, are prominent areas in need of addressing (Turner and Comiso, 2017; Li et al., 2020). The primary focus of sea ice research should thus be to establish and quantify the mechanisms responsible for driving sea ice trends, on seasonal to multi-decadal time scales.

However, research has been made challenging due to the failure of climate models to reproduce pan-Antarctic sea ice trends (Polvani and Smith, 2013; Turner et al., 2013; Roach et al., 2020). In addition, models are failing to simulate the

observed asymmetry in SIE trends between the Arctic and Antarctic regions (Rosenblum and Eisenman, 2017; Fox-Kemper et al., 2021). Far from observed trends, climate models appear to simulate a decline (although moderate) in



Antarctic sea ice extent (Rosenblum and Eisenman, 2017; Roach et al., 2020) and there is little confidence in the long-term predictions of Antarctic sea ice change from climate modelling (Fox-Kemper et al., 2021). Research is largely dependent on modelling due to the limited set of sea ice data records prior to the satellite era (Edinburgh and Day, 2016;

Fogt et al., 2022). Without accurate model simulations, our ability to quantify the primary drivers of Antarctic sea ice trends, particularly the role of anthropogenic forcing as a driving mechanism, is considerably restricted.

To assist resolution of this issue, it has been proposed that the temporal range of current sea ice records must be extended beyond the satellite era or reconstruction period (Yang et al., 2021; Fogt et al., 2022), specifically through utilising ship records (Wilkinson, 2014). Residing within many of these records is a wealth of invaluable information

regarding meteorological observations. Importantly, such information includes, but is not limited to, the nature and concentration of sea ice from observations around the globe (Wilkinson, 2014). Careful analysis of such information may provide the data necessary to enhance our understanding of the natural variability of Antarctic sea ice. In recent years there has been a concerted effort, through projects such as CLIWOC and RECLAIM, to recover meteorological data from historical records, primarily ship logbooks and meteorological registers (Konnen and Koek, 2005). These

records are particularly informative, as they contain daily observations regarding environmental conditions. Owing to projects such as CLIWOC and RECLAIM, it is possible to access data from thousands of historical records dating back to the eighteenth century, as many have now been digitised, or the data has been extracted and compiled into extensive databases (Konnen and Koek, 2005). Such databases have been made readily available to the general public. There is increasing acknowledgement that extending satellite-derived data through utilising historical records may be beneficial

not only in relation to sea ice extent, but for a range of environmental research (Kingsland, 2017).

The focus of this research has therefore been to extract and analyse information regarding sea ice observations in the relatively well-sampled Weddell Sea sector, from historical records, ultimately producing estimates for SIE during the first half of the nineteenth century. More specifically, records utilised comprise ship logbooks, meteorological registers, charts and journals, recorded during Antarctic expeditions. Information extracted from these records has been used to

estimate sea ice edge position and SIE in the Weddell Sea sector during the period 1820-43 CE. Due to the restriction of sailing era access to the Southern Ocean to the summer season, estimates will be produced only for months during the austral summer: December, January, February and March (DJFM).

## 2. Methodology

The primary aim of this project is to produce reliable estimations of summer SIE in the Weddell Sea sector, during the early nineteenth century. This information may be useful in furthering our understanding of recent sea ice trends in the Weddell Sea, specifically, how far trends are the result of internal variability, or anthropogenic forcing. To complete this aim, the methodology has been partly adapted from work conducted by Edinburgh and Day (2016). Focussing on





the Heroic Age of Antarctic exploration (1897-1917), Edinburgh and Day (2016) produced estimates for pan-Antarctic

summer SIE.

**2.1 Region and time period of study**

Whilst the completion of estimates for summer SIE on a pan-Antarctic scale is desirable, it was decided to concentrate on a restricted, but more sampled, area, thus the Weddell Sea Sector (60°W - 20°E, 76°S - 60°S) was selected as the region of study. This sector is presented in Fig. 2. Over the past century, the Weddell Sea has exhibited highly variable

sea ice trends, making it of significant interest (Yang et al., 2021; Fogt et al., 2022). In addition, following the early expeditions of explorers such as Cook and Bransfield, routes through which vessels could navigate the treacherous Southern Seas became well established, particularly within the Weddell Sea. As a result, there exists a wealth of historic records, containing data and observations for the Weddell Sea, making it an ideal region of study.

Two factors determined the selection of the time period for this project; one being the pre-existing literature, and the

other being the availability of historical records that were deemed suitable for this study, according to the project criteria (outlined in section 2.2). Very few studies have examined records from the early nineteenth century (Parkinson, 1990; Wu et al., 1999); indeed, none have done so in recent years. Given the recent effort to digitize historical marine records, there are now many more available from this time period than there were prior to 2000. Research conducted by Edinburgh and Day (2016) consisted of analysis of records from 1897-1917, so here we attempted to extend their work.

However, a number of limitations surrounding the availability of historical records further restricted the time period to 1820-43 CE. Modern-day satellite remote sensing data were used for comparison with the historical reconstructions.

**2.2 Satellite data**

The Bootstrap algorithm satellite-derived sea ice concentration dataset, obtained through NSIDC (2017) was selected for this study. This global dataset comprises daily ice concentration data recorded at a 25km resolution. Data used in

this study were available from 1978 – 2017. With a dataset of this nature it is common practice to define sea ice edge using a threshold of 15% concentration (Edinburgh and Day, 2016). The selection of this dataset was partly based on a study conducted by Worby and Comiso (2004). Their study highlighted the potential for an offset in sea ice edge position, as recorded by qualified human observers and satellite imagery. As detected via satellite imagery, the sea ice edge position was found to be approximately $0.75° \pm 0.61°$ to the south, relative to in situ observation. The Bootstrap

algorithm was chosen for this study as it delivers the greatest consistency between satellite-derived ice edge position and in situ observation (Worby and Comiso, 2004; Edinburgh and Day, 2016). However, it is important to acknowledge any errors that may exist within this dataset; therefore, this offset will be considered following the completion of statistical analysis.

**2.3 Historical data**



Nine records from eight Antarctic expeditions have been utilised in this study; the details of each record have been presented in Table 1. Specifically, these records comprise ship logbooks, meteorological registers, charts and journals. These records were selected based on their fitting of 5 criteria. These criteria are as follows:

1) Records must come from expeditions that took place, at least in part, through the Weddell Sea sector;

2) Records must contain frequent and detailed sea ice observations;

3) Records must contain regularly a log of ship position and time of day. Importantly, the meridian with respect to which ship positions were measured must be clearly stated;

4) Records must be legible in whichever format they are available;

5) Records must be accessible to the general public. In the instance of a record being inaccessible, the relevant authority must have granted permission to this record, for the purpose of this study.

Prior to completing any data extraction, an element of preparation was involved to analyse and extract relevant data efficiently. Firstly, the journal kept by Dumont D'Urville aboard the '*Astrolabe*' required translating from French to English, before any information could be extracted. A translated account of this journal is available (Rosenman, 1987) however, to include information that may have been omitted in this translated edition, the original journal was also utilised. Then, in analysing historical records, it is important to understand precisely what kind of information is held

within each record. Therefore, familiarisation with the format of ship logbooks and meteorological registers, as well as the nautical language typically found in these records, was an important first stage.

Additionally, prior to extracting information regarding sea ice observations, the various terminology used in each record was analysed and established as being representative of sea ice or not. Within each record a variety of terms are used in place of sea ice, which was only identified once in one journal account. Interpretation of these various terms can be

challenging due to the often-ambiguous way in which they are used. However, through context it was possible to deduce the meaning of different terms and thus distinguish between observations relating to sea ice and other ice forms, such as icebergs or shelf ice. In cases where the meaning can be clearly deduced, the term has been allocated its relevant meaning ('sea ice' or 'not sea ice'). A list of these terms and their classification is presented in Table 2. It should be noted that where the use of a term was ambiguous, it was classified as 'not sea ice' to keep error to a minimum.

A sea ice index was created using data extracted from each historical record. The aim of the index was to collate relevant sea ice information from each record and ultimately produce a set of summer sea ice edge position data points to be used for statistical analysis, as will be outlined in Section 2.5. From each record, where a sea ice observation was logged, five key points were noted.

These points are as follows:

1) The time of day (usually Noon) and ship's position at the time of the observation;

2) The date on which the observation was made;

3) A detailed description of the sea ice observed;



4) The term/s used to represent sea ice in the observation;

5) Any remarks made by the author, comparing their observations of sea ice with observations made on previous
expeditions.

Following the data extraction, it was necessary to make certain adjustments to make the data uniform and improve data accuracy. Firstly, longitudes logged on board the *Astrolabe* were done so with respect to the Paris Meridian, whereas all other longitudes were measured with respect to the Greenwich Meridian. The Paris Meridian is located approximately 2°20'14'' east of the Greenwich Meridian, therefore, all longitudes measured from the *Astrolabe* were adjusted
accordingly.

Next, the method of recording ship position was noted upon analysing each record. It is important to know exactly how ship positions were recorded so as to estimate the accuracy with which this was done. For example, it is well established that where the method of dead reckoning has been used, there are likely to be significant errors in the recorded longitude (Jackson et al., 2000). However, it was established that each ship position included in the sea ice index was
recorded using a chronometer, which is generally a very accurate instrument. To validate the accuracy of these positions, each route was divided into 'legs', each leg comprising a section of the route between landmarks (for which an estimated position was always noted). Then, using a British Antarctic Survey digital map of the Antarctic region, each landmark's true latitudinal and longitudinal positions were ascertained and compared to those indicated in the historical record. Where there were inaccuracies, positions along the next leg of the route were adjusted accordingly.
Perhaps surprisingly, the majority of positions were very accurate, meaning that the routes needed minimal adjustment. The only route that required significant adjustment was that of the ship *Tula*.

Following these adjustments, the data regarding sea ice observations were compiled to create the sea ice index and a set of 173 summer sea ice position data points. Of these data points, 76 were subsequently used for statistical analysis. Only data points relating to explicit sea ice observations were used for analysis. The full index is stored at the UK Polar
Data Centre at https://doi.org/10.5285/53000D3B-3069-494B-8FA9-F9C7504CAE25.

**2.3.1 Limitations of the ship-observations**

One limitation is that journal accounts contain misinformation. One journal examined for this study, that of Benjamin Morrell, who captained the *Wasp,* has been highlighted for containing inaccurate information (Stommel, 2017). Specifically, it is believed that Morrell embellished accounts of his Antarctic expedition, in relation to observations of
land sightings and the most southerly latitude to which the *Wasp* was able to sail (Stommel, 2017). Data was extracted from Morrell's journal for use in this study, yet discrepancies over the accuracy of this record call into question the reliability of the data. Examining multiple records from the same expedition may be a way to validate data to an extent, or studying a period during which several expeditions occurred, is another way to validate information. The journal accounts of James Ross, Jules Dumont D'Urville, Thaddeus Bellingshausen and James Weddell all contain details of
specific locations whereby the track of their vessel has crossed the path of a previous expedition, and often comparisons



were made regarding sea ice observations. Yet it is not always possible to do this. An additional source for error is the risk of misinterpretation or misreading of information within original historical records, or the uncertainty of whether observations referred to sea ice or something else. However, these issues have been kept in mind during data processing.

### 185 2.4 Route reconstructions

As outlined above, each record utilised in this study contains specific information regarding the time of day, latitude and longitude, noted at regular intervals. It is therefore possible, using this information, to reconstruct each route taken through the Weddell Sea. To complete this task, each ship position falling within the Weddell Sea Sector was logged in each historical record, and adjustments were made where necessary, as outlined in Section 2.3. Using Google Maps, 190 each route was then delineated onto a base map using the mapping tool. Within many of the historical records were original sketches of the expedition route with corresponding values for latitude and longitude. Therefore, these maps were used, where possible, to validate the route reconstructions. The reconstruction of the eight voyages whose data are used is shown in Fig. 2.

### 2.5 Sea ice edge latitude analysis

The initial requirement of this work was to produce estimates for summer sea ice edge position across the Weddell Sea sector, per month and decade of the nineteenth-century (1820-43) time series. These positions were then compared to those from the recent satellite (1987-2017) time series and the early twentieth-century time series comprising data obtained from Edinburgh and Day (2016). In producing estimates for summer sea ice edge position, it was then possible to produce estimates for summer SIE. The methodology associated with estimating summer sea ice edge position, as 200 adapted from that developed by Edinburgh and Day (2016), comprises three key stages and will be outlined below. The three stages are as follows:

1) Obtaining relevant satellite-derived data;

2) Isolating satellite data points for the paired analysis;

3) Paired analysis.

### 205 2.5.1 Stage 1

The satellite-derived sea ice concentration dataset comprised global daily sea ice concentration data extracted for the Weddell Sea sector. For each day of the time series, the average ice edge position was calculated, this being defined as the contour connecting the northernmost pixels, with a sea ice concentration below 15%. Each pixel represents an area of $25km^2$, however, to reduce the number of latitudinal data points that comprised each contour, the data was averaged 210 over 2° longitude bands across the Weddell Sea.

### 2.5.2 Stage 2

In order to conduct the paired analysis, a number of stages were first required. For each historical ship-observed sea ice position, a satellite-derived sea ice edge latitude value for the corresponding day, for each year within the satellite time



series was obtained. This involved checking, for each longitude value and taking into account the sphericity of the
Earth, the distance to the ship-observed position and selecting the point closest in geographical distance for that specific
day. The latitude with the smallest distance from the ship-observed position was thus identified. The result was a set of
30 latitudinal points (as the satellite time series spans 30 years) corresponding to each ship-observed position. At the
end of this stage, a set of satellite-derived ice edge latitudes had been calculated, corresponding to the relevant position
and time of year, of each of the 76 historical ship-observed positions. These sets of latitudinal values were then used for
the paired analysis, as will be outlined next.

### 2.5.3 Stage 3

The final stage was to conduct paired analysis between the historical ship-observed ice edge positions and each of their
corresponding 30 satellite-derived ice edge positions. The Wilcoxon signed-rank test was the selected form of statistical
testing, as this does not assume normal distribution of the data and allows for comparison of means within each data set.
Ultimately, each ship-observed position was analysed against its 30 satellite-derived ice edge latitude points. The 30
results were then averaged for each ship-observed position and grouped by decade and month of the study period.

Taking the same approach, data from the nineteenth century time series were also compared to data obtained from
Edinburgh and Day (2016), for the Weddell Sea sector. Edinburgh and Day (2016) completed pan-Antarctic estimates
for summer sea ice edge latitude during the period 1897-1917, therefore their data set for only the Weddell Sea is fairly
limited. Specifically, the data set comprises 15 ship-observed data points and they possess no data for sea ice
observations during December or January. Despite this, comparison between the time periods allows for a more holistic
examination of summer sea ice trends.

### 2.6 Sea ice extent analysis

An estimate of the Weddell Sea SIE was also made from the historical ship observations. For each historical ship
observation, the mean sea ice edge latitude difference - calculated from the paired analysis - was assumed to occur
across the mean sea ice edge boundary of the Weddell basin for that day of the year, leading to a first order estimate for
the change in SIE between the modern era and the date of each ship-observed sea ice position. By combining the values
for each month, or each decade, first order estimates of mean SIE change between the early nineteenth century and
today can be calculated.

### 3. Results

### 3.1 Data available per month and decade

The bar graphs presented in Fig. 3 illustrate the number of ship-observed sea ice positions recorded during each month
and decade of the nineteenth century time series. Fig. 3a presents the number of observations made per month. The
highest numbers of ice edge positions were recorded during January (43 values), followed by February (16 values), then
December (11 values) and the lowest number of positions was recorded during March (6 values). Fig. 3b presents the
number of observations made per decade, including the number of those observations recorded during each given





month. The highest number of ship-observed ice edge positions were recorded during the 1840s (47 values), including observations from January, February and March. During the 1820s 21 positions were recorded, including observations from all months. Finally, the lowest number of ship-observed positions were recorded during the 1830s (8 values),

including observations from January and February only.

### 3.2 Sea ice edge comparison: 1820-43 with modern

Table 3 and Fig. 4 present the results of the paired analysis, for comparison of summer sea ice edge position between the nineteenth century and modern time series, per decade. Table 3 shows that on average, summer ice edge position was further north in each decade of the nineteenth-century time series, relative to today. All results were significant at

the 95% confidence level. The greatest difference in summer ice edge latitude between the two time series occurred during the 1830s, although there were far fewer observations in that decade. Fig. 4 illustrates the position of each ship-observed ice edge point, in relation to the average satellite-derived ice edge position. Mostly, the ship-observed ice edge data points fall north of the satellite-derived ice edge, as expected, given the paired analysis results (Table 3). However, this is not consistently so, for either decade or longitude, so it is instructive to examine monthly comparisons to explore

the historical contrast further.

Table 4 and Fig. 5 present the results of the paired analysis, for comparison of summer sea ice edge position between the satellite and historical time series, per month of the historical time series (DJFM). Results in Table 4 show that during December and March the nineteenth century ice edge latitude was essentially similar to those of the modern time series across the Weddell Sea. However, during January and February the ice edge in the nineteenth century time series

was further north by a statistically significant 0.16° and 2.36°, respectively. However, Fig. 5 shows the ship-observed ice edge data points that fall north of the satellite-derived ice edge in January and February are essentially from the central part of the Weddell Sea, rather than its western or eastern peripheries, meaning that in the core Weddell Sea the sea ice melts much slower in the nineteenth century decades than today.

### 3.3 Sea ice edge comparison: 1820-43 with 1897-1917

There is a limited amount of ship-observed sea ice latitude data from the Weddell Sea in the Heroic Age, and it is from only 3 years, some of which may have been exceptionally ice-heavy due to a simultaneous El Niño event (Edinburgh and Day, 2016). However, Edinburgh and Day (2016) found the Weddell Sea sector sea ice to be significantly further north than during the satellite era; this is also the case when comparing with the observations considered here. Only the months of February and March have sufficient data to be comparable but the Heroic Age sea ice latitude is significantly

further north than either the 1820s-1840s or the modern era (Table 5).

### 3.4 Sea ice extent comparison: 1820-43 with modern

The sea ice extent (SIE) estimates presented here, whose methodology was explained in section 2.6, are approximate and rely on local, one-day differences being representative of that day for the whole Weddell Sea sector. However, they are generally consistent with the sea ice edge latitude analysis. For the months of December-March, only the relatively



well-observed month of February had a nineteenth century SIE greater than today's mean, while the other nineteenth century summer months had a similar SIE to today (Table 6). Nevertheless, all three decades of the nineteenth century time series had marginally to significantly greater SIEs, on average, than today (Table 7).

**4. Discussion**

The paired analysis of section 3.2 indicates that, per decade, all ice edge latitudes were further north during the
nineteenth century, relative to the satellite era. However, Table 3 suggests the 1820s had more extensive sea ice than either of the other decades studied. Furthermore, the inclusion of the Worby and Comiso (2004) offset does not impact the outcome of these results; i.e. on average the summer sea ice edge position remains further north in the nineteenth century series, relative to the satellite era time series.

Interestingly, the mean satellite-derived ice edge latitudes display a similar decadal trend, as the difference between
ship-observed and satellite ice edges remains roughly the same over time. Between satellite data corresponding to data from the 1820s and that corresponding to the 1830s, there is a retreat of the ice edge latitude of 3.55°. Between satellite data corresponding to data from the 1830s and that corresponding to the 1840s, there is an advance in ice edge latitude of 1.48°. It should be kept in mind that when conducting this analysis, only values from the satellite-derived dataset from days corresponding to historical observations were included. It is perhaps also important to note that within the
historical dataset, ship-observed ice edge values during the 1830s were only recorded during January and February. Within the austral summer, the greatest retreats in ice edge would be expected to occur during these months, particularly February (Turner et al., 2013). However, Table 4 suggests the largest sea ice retreat in the first half of the nineteenth century occurs in the late austral summer, with March showing the furthest south ship-observed sea ice latitude. It needs to be noted, however, that the lowest number of ship-observed data points were obtained during March
(Fig. 5), although these match well the satellite-era sea ice positions near the few observed longitudes.

Our comparison with the limited Weddell Sea Heroic Age data suggested that the nineteenth century ice edge latitude was further south during both February and March, the two months with Heroic Age data. Variability on quasi-quadrennial to inter-decadal scales is typical within the Weddell Sea (Gloersen et al., 1993); therefore, these comparisons over almost two hundred years, from the 1820s through 1900 to the 2010s, do not indicate any clear trend
from the underlying inherent decadal variability.

Today the minimum SIE is typically observed in February (Turner et al., 2013). However, the monthly average SIE estimates produced in this study show the greatest nineteenth century monthly extent to occur during February, with the minimum most likely to be in March. It is difficult to interpret this result without obtaining additional data to validate the estimation. This being said, a number of studies have demonstrated monthly variability in SIE on a range of broader
temporal scales. Parkinson (1992) and Morioka et al. (2017) have demonstrated inter-annual monthly variability in SIE. De Santis et al. (2017) demonstrated inter-decadal monthly variability in SIE. Furthermore, Turner et al. (2013) identified an increase in SIE during February and March, in model simulations run with historical forcing (1850s-2005).



Thus, the nineteenth century average SIE may simply be part of the variability in monthly average SIE, on broader temporal scales.

**5. Conclusions**

This analysis has shown that while the period 1820-43 experienced relatively more sea ice in the Weddell Sea than in the satellite era, the general tendency for Weddell Sea summer sea ice extent over the last two hundred years is dominated by decadal variability, with other periods likely experiencing even more sea ice than the first half of the nineteenth century (Table 5). This is consistent with several other studies of the twentieth century alone (Murphy et al., 320 2014; Yang et al., 2021; Fogt et al., 2022). Nevertheless, both Edinburgh and Day (2016) in their study of the Heroic Age and the present nineteenth century study suggest the seasonal sea ice minimum formerly occurred later in the austral summer compared to today.

There are limitations in this analysis, as has been discussed in sections 2 and 3, principally related to the interpretation of the journal entries and more particularly to the limited, and seasonally varying, number of ship observations of sea 325 ice edge. Much relies on using the modern record for the days of the year actually observed in the Weddell Sea, in 1820-43. Nevertheless, the often statistically significant results, and the consistency with the decadal variability tendency shown by other studies, gives confidence in the primary outcome of the analysis of a more extensive Weddell Sea sea ice cover in the nineteenth-century, and a later seasonal minimum, compared to today. Further study will be made in the future of pan-Antarctic voyages in order to give general Southern Ocean sea ice reconstructions of the 330 period.

**Data availability.** The full dataset of transcribed nineteenth-century documentary references in the source logbooks and journals is available through the UK Polar Data Centre at https://doi.org/10.31223/X5M07N.

**Author contributions.** The document transcribing and statistical analysis was carried out by EL, under the supervision of GRB, for a dissertation as part of EL's Masters in Geography. The text was written by both EL and GRB, with the 335 diagrams and tables used in the paper prepared by GRB.

**Competing interests.** The contact author has declared that neither they nor their co-author have any competing interests.

**Acknowledgements.** The authors would like to thank Théo Michelot for his help in translating the journal of Dumont D'Urville. The authors would also like to acknowledge the Bartlett Library at the Royal Maritime Museum, Falmouth, 340 UK for generously donating the use of their microfilm reader and for their support in enabling an international library loan with the New Bedford Whaling Museum, Massachusetts. Much of this work formed the Masters dissertation of EL.

**Financial Support.** The preparation and publication of this paper was funded by a Leverhulme Trust Emeritus Fellowship.




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





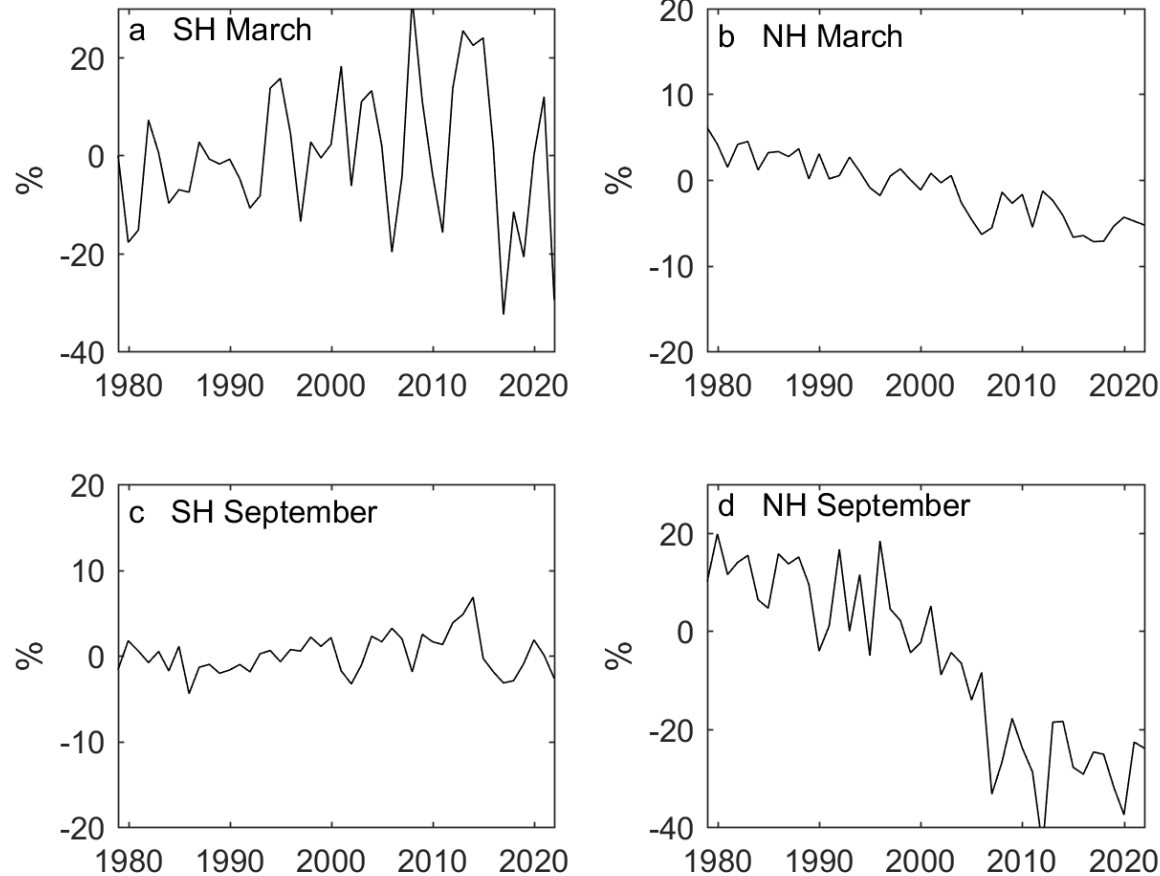

**Figure 1: Antarctic (a summer, c winter) and Arctic (b winter, d summer) sea ice extent over 1979-2022, relative to the 1981-2010 means.**



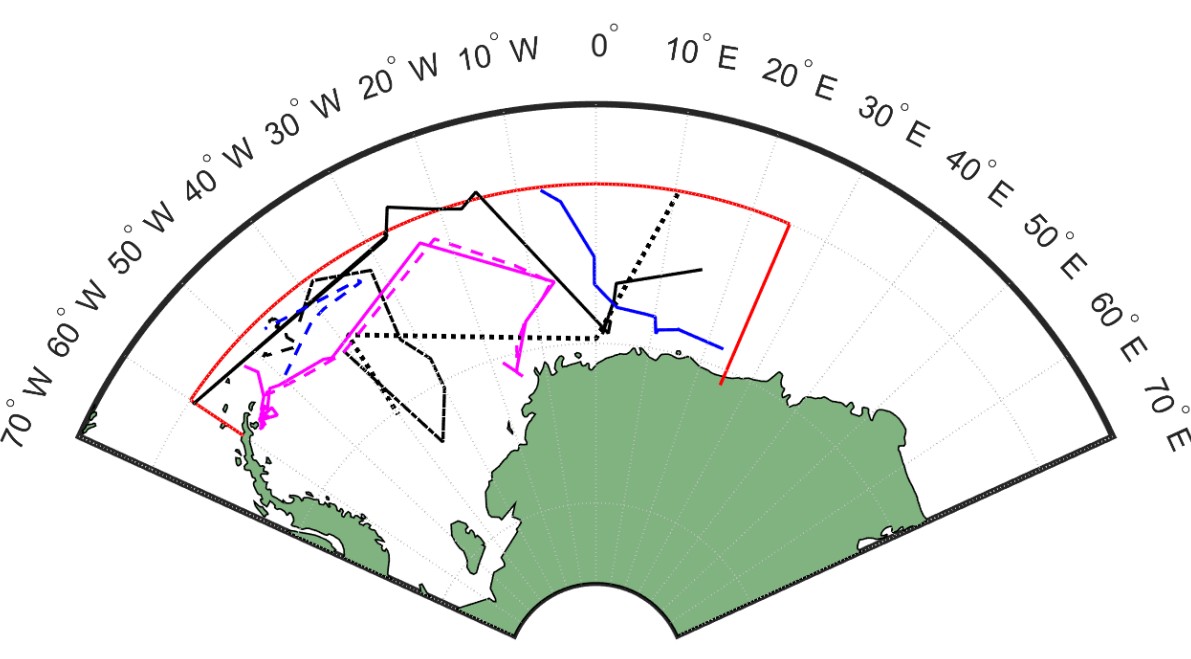

**Figure 2: Map showing region of interest in Weddell Sea (outlined by red lines) and the paths of the various vessels whose data is used here. Black lines date from the 1820s (solid – Bellinghausen; dashed – Powell; dotted – Morrell; dot-dash – Weddell), blue from the 1830s (solid – Biscoe; dashed – Dumont) and magenta from the 1840s (solid – Crozier; dashed – Ross).**



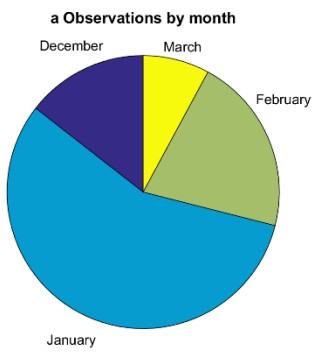

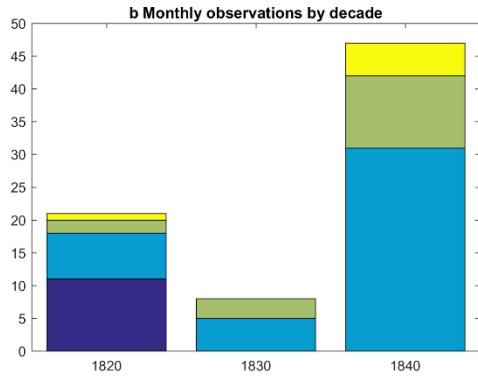

**Figure 3: Nineteenth-century observations by month: a) monthly proportion; b monthly observations by decade.**






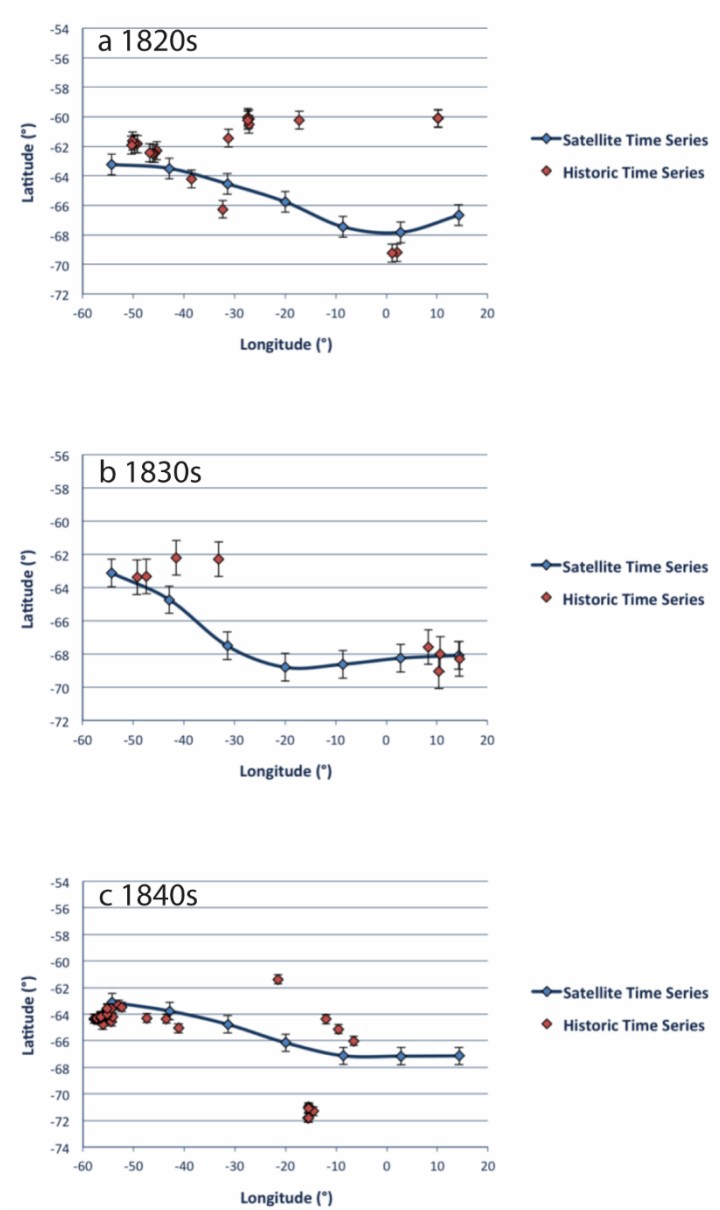

**Figure 4: Decadal comparison of satellite time series with nineteenth-century data; a) 1820s; b) 1830s; c) 1840s. Satellite data error bars show standard deviation for sea ice latitude at times of the year with nineteenth century data. Historic data error bars give an estimate of the relative accuracy of latitude location each decade.**






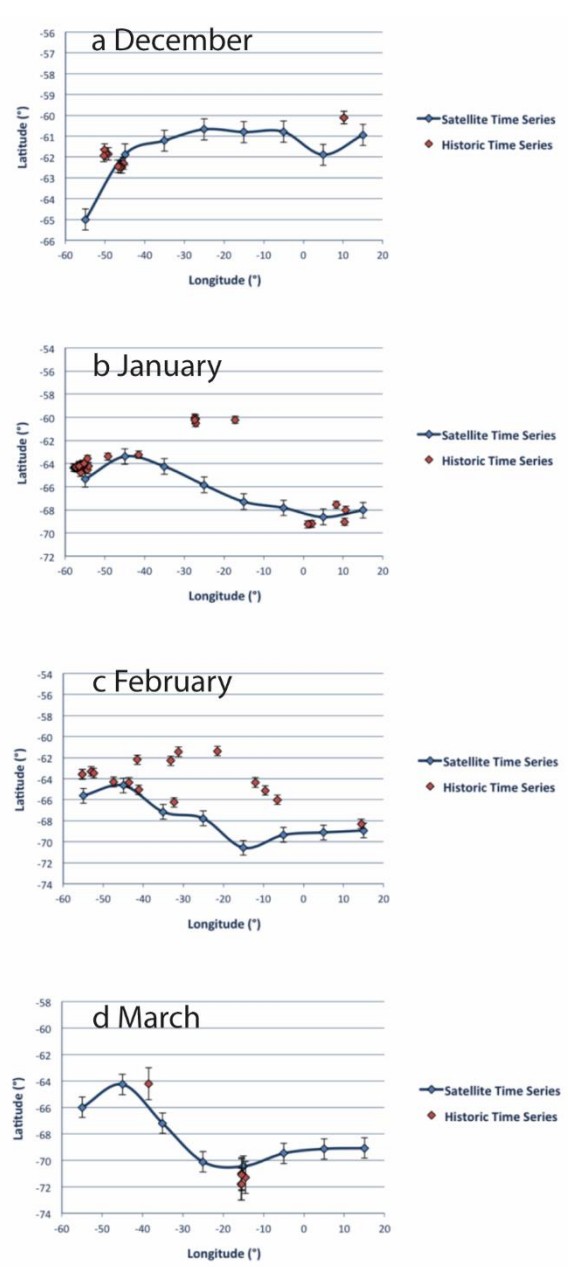

**Figure 5: Monthly comparison of satellite time series with nineteenth-century data; a) December; b) January; c) February; d) March. Satellite data error bars show standard deviation for sea ice latitude at times of the year with nineteenth century data. Historic data error bars give an estimate of the relative accuracy of latitude location each decade.**



**Table 1.** Details of the historical records used in this study (CEDA: Centre for Environmental Data Analysis). The "ship" column also includes the expedition number for this study. The Bellingshausen voyage is available in translation (Debenham, 2010) as well as the D'Urville voyage (Rosenman, 1987).

| Date | Source | Author | Nature of Voyage | Ship | Place of Access |
|---|---|---|---|---|---|
| 1819-21 | Journal | Bellingshausen | Exploration | *Vostok* (1) | Archive.org |
| 1822-32 | Journal | Benjamin Morrell (1832) | Sealing | *Wasp* (2) | Archive.org |
| 1821-22 | Journal | Captain Powell | Sealing | *Dove* (3) | National Archives |
| 1822-24 | Journal | James Weddell (1834) | Exploration | *Jane* (4) | Archive.org |
| 1830-33 | Journal | Biscoe and Enderby (1833) | Sealing | *Tula* (5) | Archive.org |
| 1837-40 | Journal | Jules S-C Dumont D'Urville (1842) | Exploration | *Astrolabe* (6) | Archive.org |
| 1839-43 | Journal | James Ross (1847) | Exploration | *Erebus* (7) | Archive.org |
| 1839-43 | Logbook | C F Tucker | Exploration | *Erebus* (7) | CEDA (2011a) |
| 1839-43 | Logbook | Francis Crozier | Exploration | *Terror* (8) | CEDA (2011b) |





**Table 2.** Terms identified in historic records and their classification as representing sea ice or not. Y=Yes (sea ice), N=No (not sea ice). The 'Expedition' column in this table refers to the expedition from which records containing each
terminology originated; labelled from 1-8 in the key below is each expedition (see Table 1 for key).

| Term | Sea Ice Classification | Expedition |
|---|---|---|
| Field ice | Y | 1, 2, 3, 5, 6, 8 |
| Heavy ice | Y | 4, 7, 8 |
| Compact ice | Y | 5 |
| Close pack | Y | 7, 8 |
| Pack ice | Y | 1, 4, 7, 8 |
| Loose pack ice | Y | 8 |
| Drift ice | Y | 2, 3, 5 |
| Ice Floe | Y | 1, 7 |
| Sea ice | Y | 6 |
| Pancake ice | Y | 1 |
| Low ice | Y | 1 |
| Ice field | Y | 1 |
| Young ice | Y | 7 |
| Floating ice | Y | 1, 3, 5 |
| Hummocky ice | Y | 1 |
| Black ice | Y | 4 |
| Broken ice | N | 1, 5, 8 |
| Ice hillocks | N | 1 |
| Straggling ice | N | 5 |
| Loose ice | N | 5, 7, 8 |
| Fallen ice | N | 1 |
| Small ice | N | 1, 2 |
| Stream of ice | N | 7 |
| Open water | N | 3, 4, 5, 7, 8 |
| Ice island | N | 4, 5, 6 |
| Barrier ice | N | 6 |



**Table 3.** Paired analysis results, comparing summer ice edge latitude between the relevant days from the modern, satellite, time series and each decade of the nineteenth-century time series. In the mean difference row positive numbers represent degrees north from today; values significant at 95% level in **bold**.

|  | 1820s | 1830s | 1840s |
| --- | --- | --- | --- |
| Mean ship-observed IEL (ºS) | 62.32±2.70 | 65.5±2.78 | 64.96±2.30 |
| Mean difference with today (º) | **0.7** | **1.07** | **0.13** |

**Table 4.** Paired analysis results, comparing summer ice edge latitude between the modern time series and each month of the nineteenth-century time series. In the mean difference row positive numbers represent degrees north from today; values significant at 95% level in **bold**.

|  | December | January | February | March |
| --- | --- | --- | --- | --- |
| Mean ship-observed IEL (ºS) | 61.57±0.94 | 64.24±2.05 | 64.06±1.79 | 70.2±2.70 |
| Mean difference with today (º) | -0.02 | **0.16** | **2.36** | -0.2 |

**Table 5.** Paired analysis results, comparing summer ice edge latitude during the months of February and March, between the nineteenth and twentieth century time series. In the mean difference row positive numbers represent degrees north from the Heroic Age (1897-1917).

|  | February | March |
| --- | --- | --- |
| Mean ship-observed IEL (ºS) | 64.06±1.79 | 70.2±2.70 |
| Mean difference with Edinburgh and Day (2016) (º) | -0.85 | -1.71 |

**Table 6.** Sea ice extent estimations for each month of the nineteenth-century time series. Also presented are estimated differences between these values and sea ice extent averaged over the satellite period.

|  | December | January | February | March |
| --- | --- | --- | --- | --- |
| Mean SIE ($10^6$ km$^2$) | 11.454 | 4.931 | 5.904 | 3.278 |
| Mean Difference in SIE ($10^6$ km$^2$) | -0.029 | -0.018 | 2.473 | -0.195 |

**Table 7.** Sea ice extent estimations for each decade of the nineteenth-century time series. Also presented are estimated differences between these values and sea ice extent averaged over the satellite period.

|  | 1820's | 1830's | 1840's |
| --- | --- | --- | --- |
| Mean SIE ($10^6$ km$^2$) | 5.359 | 6.083 | 4.072 |
| Mean Difference in SIE ($10^6$ km$^2$) | 0.362 | 0.845 | 0.124 |