# Peer review of "Estimating summer sea ice extent in the Weddell Sea during the early nineteenth century"

_Climate of the Past, 2023_

## Referee Comment (RC1)

Title: Estimating summer sea ice extent in the Weddell Sea during the early nineteenth century
Author(s): Eleanor Love and Grant R. Bigg
MS No.: cp-2023-4
MS type: Research article
Iteration: Initial submission
Referee: Seelye Martin

General Comments: This is an interesting paper that with some corrections and additions, should be suitable for publication. As its abstract states, it makes novel use of ship data from the 18[th] century in an examination of summer sea-ice positions in the Weddell Sea from nine traverses derived from eight Antarctic expeditions during 1820-1840. It compares these with the Comiso bootstrap passive microwave satellite observatiions from 1987-2017, plus historical data from 1987-2017. Its key finding is that in the nineteenth century, the summer sea ice latitude in much of the Weddell Sea was much further north than during the satellite era.

This reviewer was greatly impressed that in the comparison, and as described on line 214, the satellite data compared with each ship point was the average for the same date, over each of the satellite years. Marvellous!

Also, as described in point 3 below, I strongly urge the authors to consider the addition of Cook's Weddell sea-ice data from 1772-73, as described in their Parkinson (1990) citation. This data is published in an Excel format in the Martin paper below, so its addition should be relatively easy.

Specific Comments: There are three problems that the authors need to address:

1. In their Figure 2, for the 1820s, why does the black line, Bellinghausen, extend north of the region of red-line region of interest? Also, this reviewer cannot distinguish the dotted from the dot-dash line. Perhaps you should use a separate color to distinguish these two. Also, Parkinson (1990, Table I) defines the Weddell as extending to 30°E, to include all of Cook's observations. Do you want to extend it to match her discussion?

2. For December in the Weddell, which has multiple regions of ice and open water, how do we know what the ice edge represents? For example, the figure below is Figure 3 from the recently published open access paper by Martin and others (2022, reference on next page).

[Figure]

Fig. 3. Passive satellite image of the Antarctic sea-ice extent for 14 December 2018. Sea ice is white, water is light gray, the continent is dark gray. For the same date and the thirty-year period 1981–2010, the black lines show the median ice extent. The open water associated with the Maud Rise Polynya (indicated with arrow) is visible in both the daily and median image (Image courtesy NSIDC).

Martin S, Long DG, Schodlok MP (2022). Comparison of Antarctic iceberg observations by Cook in 1772–75, Halley in 1700, Bouvet in 1739 and Riou in 1789 with modern data. Journal of Glaciology 1–8. https://doi.org/10.1017/jog.2022.111

Question for authors: for this date, where would you put the ice edge in the Weddell? Following your discussion in your Section 2.5, how do you handle observations of the multiple ice edges as shown in this figure?

This is not an academic question. As Parkinson (1990) describes, and Martin and others (2022) show, Cook mapped the peninsula-like feature in this figure (the more southerly feature just above the polynya). Parkinson (1990) discusses the peninsula and shows that, compared to a similar feature in the 1973-1976 passive microwave data, it occurs several weeks later, which she suggests may be the result of a colder climate. She shows that his 14–18 December observations in the vicinity of 55°S are 2–8° farther north than the mid-December satellite ice edge at the same longitudes in 1973–76.

3. Following Parkinson, consider addition of Cook's data to your analysis. As stated above, the Supplementary Material in Martin and others (2022) contains Cook's observations of icebergs and the ice edge in an Excel format, which can be easily uploaded into Google Earth. Since Cook's first traverse in December 1772 delineates the sea-ice peninsula shown in Fig 3, this would make a excellent addition to the authors' work.

Parkinson concludes that " the evidence from this preliminary study suggests that it is unlikely that a thorough study will yield a strong and consistent Little Ice Age signal from the sea ice of the Southern Ocean, at least for the 1770–1850 period." Do the authors agree with this? Your analysis combined with Parkinson's suggests that for the 1770–1850 period, the Dec-Jan-Feb ice cover in the Weddell is enhanced.

Given that Cook's data is now formatted and easily available, its addition to their paper would consume a negligible amount of time, enhance their citation of Parkinson (1990), extend their analysis fifty years back in time to 1772-73, give them more data east of the prime meridian in the December Weddell, and allow them to comment on the historic regional climate. For these reasons, I hope the authors extend their analysis through addition of Cook's data.

Techical Corrections:

General: Would like to have **every** line numbered, not just 5, 10, 15, 20…

Line 73: The following sentence needs work."More specifically, records utilised  ship logbooks, meteorological registers, charts and journals,  [from the various] Antarctic expeditions."

Line 89: Rewrite the end of the first sentence, which currently reads "region of study. This sector is presented in Fig. 2." As "…region of study (Fig. 2)." This saves five words.

Line 96: point out that Parkinson (1990), in her Little Ice Age paper, also examined Cook's Weddell data from 1772-73, as well as data from the early nineteenth century.

Line 145: Time and date of observation. Are you sure that all of your ships are using civil time and the civil callendar for the dates? Or, are some of them still using naval time (noon-to-noon for days)?

Line 160: How accurate in terms of km or nm is the longitude determined from a chronometer?

Line 338: Minor edit in Acknowledgements. "The authors  thank Théo…"

Figures: note with the possible exception of Figure 3, the figures are necessary.

---

## Referee Comment (RC2)

Initial review of cp-2023-4: Estimating summer sea ice extent in the Weddell Sea during the early nineteenth century

General Comments:

The manuscript focusses on extending the temporal range of sea ice records beyond the satellite era or reconstruction period, through digitization of past records. The presented sea ice extent (SIE) estimates from this study seems important to understand the links between past and present SIE trend of Western Antarctica.

The figure and illustrations of the observed and satellite-based interpretation could be improved. The document could be corrected for repetition and long sentences that are compromising clarity. Overall, the manuscript can be accepted for publication after incorporating minor changes as suggested in the specific comments below.

Specific Comments:

The authors mainly rely on the ship records to recreate past SIE and have talked about the inaccuracies to some extent (sec. 2.3). To clearly understand the underlying inaccuracies, the authors should state the uncertainties in the reference British Antarctic Survey digital map of the study region. Were any checks in digitization accuracy applied for the region during 19th century? Also, it would be useful to let readers know about the landmark's used in the study to ascertain that the landmarks have not changed in the 200 years period. The errors in the route are evaluated 'relatively' and all the ship measurements were taken using chronometer. What was the reason that only ship Tula required significant adjustment? How big these corrections were? Stating this here might be of interest to researchers handling similar datasets.

The manuscript presents a significant amount of detail in observed data preparation which is commendable. However, pictorial illustrations of the records seem to be missing. For e.g., Fig. 2 should be updated with a background imagery (possibly optical satellite imagery) to reflect the different sea ice features present in the study region. The manuscript aims to 'estimate summer sea ice extent in the Weddell Sea during the early nineteenth century' and a comparative trend analysis has been discussed and presented. However, a concluding illustration is missing. The reviewer suggests adding a map (along with a proper base map) showing an appropriate combination of 1. the latitudinal locations of ice edges identified, 2. corresponding satellite dataset for selected timestamp, and 3. a final quantified SIE extent (with total estimated area in $km^2$).

Figure 3a seems unnecessary. The caption for Figure 3 can be more descriptive. For e.g., Nineteenth-century observations of what? The number of total observations recorded is least for March and

highest for January. However, December observations are only available for one decade i.e., 1820. Did the authors investigate any effect due to this? The authors need to state this clearly in the discussions as it affects the interpretation of multidecadal DJFM analysis specially while comparing with other studies.

Technical Corrections:

The manuscript can be improved by correcting for some minor changes:

Line 200: delete 'and will be outlined below '. It is repetitive with the successive sentence.

Fig 2: All the labels are not mentioned in the caption.

Line 192-193: 'The reconstruction … Fig. 2' can be simply written as 'The reconstructed voyages' data is shown in Fig. 2.'

---

## Author Response (AR2)

Reply to Referee Seelye Martin's comments

Firstly, we would like to thank Dr. Martin for his helpful comments on our paper, as well as his thoughts on improvements. One of the main points raised by Dr. Martin was the possible addition of Cook's data to our study. We argued in the Discussion, which was agreed by Dr. Martin, that we plan a near-future publication comparing 18[th] and 19[th] century data around Antarctica and this would be the appropriate place for such a discussion. In this regard, GRB has already compiled data from both Cook and Furneaux's journals for the 1770s expedition. We note our response to the 3 main comments and technical points below:

*Reviewer Point 1*

*1. In their Figure 2, for the 1820s, why does the black line, Bellinghausen, extend north of the region of red-line region of interest? Also, this reviewer cannot distinguish the dotted from the dot-dash line. Perhaps you should use a separate color to distinguish these two. Also, Parkinson (1990, Table I) defines the Weddell as extending to 30°E, to include all of Cook's observations. Do you want to extend it to match her discussion?*

Authors' Response:

We have improved the line marking/colouring in the final figure. Our chosen Weddell Sea area, shown by the black line in Figure 1, is only indicative. Bellinghausen's track basically determined our northern extent, but this wobbled back and forward around 60S. We now note this in the revision. We have altered the text to refer more fully in the Introduction to Parkinson (1990), as well as Martin et al. (2022) and Martin (2023).

*Reviewer Point 2*

*2. For December in the Weddell, which has multiple regions of ice and open water, how do we know what the ice edge represents?*

*Question for authors: for this date, where would you put the ice edge in the Weddell? Following your discussion in your Section 2.5, how do you handle observations of the multiple ice edges as shown in this figure?*

Authors' Response:

The reviewer raises the very good point here that in places the sea-ice in the Weddell Sea is not continuous, but includes polynyas and eastward extensions at its northern end that have open water regions to the south. In our paper we are dealing with the northern limit of sea-ice, on the assumption that even if open water existed further south this would not be detectable by the explorers. We have made this clearer in the revision and also added an inset to Fig. 2 to show an example of sea-ice where the northern limit does not infer complete ice cover to the Antarctic coast..

*Reviewer Point 3*

*3. Following Parkinson, consider addition of Cook's data to your analysis. As stated above, the Supplementary Material in Martin and others (2022) contains Cook's observations of icebergs and the ice edge in an Excel format, which can be easily uploaded into Google Earth. Since Cook's first traverse in December 1772 delineates the sea-ice peninsula shown in Fig 3, this would make a excellent addition to the authors' work.*
*Parkinson concludes that " the evidence from this preliminary study suggests that it is unlikely that*

*a thorough study will yield a strong and consistent Little Ice Age signal from the sea ice of the Southern Ocean, at least for the 1770–1850 period." Do the authors agree with this? Your analysis combined with Parkinson's suggests that for the 1770–1850 period, the Dec-Jan-Feb ice cover in the Weddell is enhanced. Given that Cook's data is now formatted and easily available, its addition to their paper would consume a negligible amount of time, enhance their citation of Parkinson (1990), extend their analysis fifty years back in time to 1772-73, give them more data east of the prime meridian in the December Weddell, and allow them to comment on the historic regional climate. For these reasons, I hope the authors extend their analysis through addition of Cook's data.*

Authors' Response:

We have already addressed this point in the Discussion, we think to Dr. Martin's satisfaction. We feel that the current paper is looking at the decadal variability of a specific 30 year period and that adding data from 50 years previously would distort the paper's focus; this focus on the 19th century is made clearer in the revision text.

*Reviewer's Technical Corrections:*

*General: Would like to have **every** line numbered, not just 5, 10, 15, 20…*
*Line 73: The following sentence needs work."More specifically, records utilised comprise ship logbooks, meteorological registers, charts and journals, recorded during [from the various] Antarctic expeditions."*
*Line 89: Rewrite the end of the first sentence, which currently reads "region of study. This sector is presented in Fig. 2." As "…region of study (Fig. 2)." This saves five words.*
*Line 96: point out that Parkinson (1990), in her Little Ice Age paper, also examined Cook's Weddell data from 1772-73, as well as data from the early nineteenth century.*
*Line 145: Time and date of observation. Are you sure that all of your ships are using civil time and the civil calendar for the dates? Or, are some of them still using naval time (noon-to-noon for days)?*
*Line 160: How accurate in terms of km or nm is the longitude determined from a chronometer?*
*Line 338: Minor edit in Acknowledgements. "The authors would like to thank Théo…"*
*Figures: note with the possible exception of Figure 3, the figures are necessary.*

Authors' Response:

We are happy to adjust the manuscript lines 73, 89, 96 and 338 as suggested by the reviewer and have done this in the revision. Thank you for pointing these out. Line 145: The ships were using civil time and positions are usually at midday on the date given. This is noted in the metafile on the UKPDC entry. Line 160: We have added the accuracy estimate for the longitude value as noted in Sobel (1996) paper of 6-19 km in the revision of the manuscript. Figure 3a has been removed, as much of the information is in the text.

Reply to Referee Two's comments

Firstly, we would like to thank the reviewer for their generally positive impression of our paper. We have checked the manuscript so as to remove unnecessary repetition and over-long sentences in the revised version. We note our response to the 3 main comments and technical points below:

*Reviewer Point 1*

*The authors mainly rely on the ship records to recreate past SIE and have talked about the inaccuracies to some extent (sec. 2.3). To clearly understand the underlying inaccuracies, the authors should state the uncertainties in the reference British Antarctic Survey digital map of the study region. Were any checks in digitization accuracy applied for the region during 19th century? Also, it would be useful to let readers know about the landmark's used in the study to ascertain that the landmarks have not changed in the 200 years period. The errors in the route are evaluated 'relatively' and all the ship measurements were taken using chronometer. What was the reason that only ship*

*Tula required significant adjustment? How big these corrections were? Stating this here might be of interest to researchers handling similar datasets.*

Authors' Response:

The BAS digital map was assumed to be highly accurate for the coastal regions being considered, with any errors under 1 km. As far as we are aware there will have been no significant changes in the landmarks chosen by the ships for navigation purposes since the nineteenth century, as these were islands or other coastal landmarks, and not glaciers which might have changed position over time. As noted in the reply to Reviewer 1, we have added the accuracy estimate for the chronometer-determined longitude value as noted in Sobel (1996) of 6-19 km in the revision of the manuscript. Almost all of the vessels whose journals were used were naval vessels, with similar levels of positional accuracy independent of nation. The Tula was a sealing vessel from the 1830s, and so may have had less reason for the accuracy required of naval captains. There was no indication from the journal as to whether the errors were instrument-related or just human calculation error. The 1820s sealing cruises were often manned by crews with Napoleonic war-time experience and so probably higher levels of experience and did not require adjustments to their positions.

*Reviewer Point 2*

*The manuscript presents a significant amount of detail in observed data preparation which is commendable. However, pictorial illustrations of the records seem to be missing. For e.g., Fig. 2 should be updated with a background imagery (possibly optical satellite imagery) to reflect the different sea ice features present in the study region. The manuscript aims to 'estimate summer sea ice extent in the Weddell Sea during the early nineteenth century' and a comparative trend analysis has been discussed and presented. However, a concluding illustration is missing. The reviewer suggests adding a map (along with a proper base map) showing an appropriate combination of 1. the latitudinal locations of ice edges identified, 2. corresponding satellite dataset for selected timestamp, and 3. a final quantified SIE extent (with total estimated area in km$_2$).*

Authors' Response:

We have considered this issue more in the light of carrying out the revision. We have modified Figure 2 to improve the clarity of the Figure and have added an inset, showing a typical December sea ice map. We considered adding a summary figure, however, Tables 6 and 7 give as much information as is realistic for the SIE estimate. Figure 5 shows how the reconstruction compares in a geographical sense to the satellite data in a very clear way. In combination, along with some additional text in the Discussion section on the SIE estimates, we feel we give sufficient information n the revision to obtain a view of the 19[th] century SIE compared to present day, without using either an overly confusing, or too simple, summary figure..

*Reviewer Point 3*

*Figure 3a seems unnecessary. The caption for Figure 3 can be more descriptive. For e.g., Nineteenth-century observations of what? The number of total observations recorded is least for March and highest for January. However, December observations are only available for one decade i.e., 1820. Did the authors investigate any effect due to this? The authors need to state this clearly in the discussions as it affects the interpretation of multidecadal DJFM analysis specially while comparing with other studies.*

Authors' Response:

We agree with the reviewer and have removed Figure 3a in the revision, as well as modifying the relevant text.

*Reviewer's Technical Corrections:*

*Line 200: delete 'and will be outlined below '. It is repetitive with the successive sentence.*
*Fig 2: All the labels are not mentioned in the caption.*
*Line 192-193: 'The reconstruction … Fig. 2' can be simply written as 'The reconstructed voyages' data is shown in Fig. 2.'*

Authors' Response:

We have adjusted line 200 as suggested.

Figure 2 has been revised and the Figure legend extended.

We have adjusted line 192-3 as suggested.